# Evaluation of Pharmacobezoar Formation from Suspensions of Spray-Dried Amorphous Solid Dispersions: An MRI Study in Rats

**DOI:** 10.3390/pharmaceutics15030887

**Published:** 2023-03-09

**Authors:** Hannes Gierke, Susan Mouchantat, Sabine Berg, Michael Grimm, Stefan Hadlich, Marie-Luise Kromrey, Thomas Nolte, Teresa Pfrommer, Vincent Rönnpagel, Adrian Rump, Kerstin Schaefer, Ann-Cathrin Willmann, Werner Weitschies

**Affiliations:** 1Department of Biopharmaceutics and Pharmaceutical Technology, Center of Drug Absorption and Transport, University of Greifswald, 17489 Greifswald, Germany; 2Department of Diagnostic Radiology and Neuroradiology, University Medicine Greifswald, 17475 Greifswald, Germany; 3Central Core & Research Facility of Laboratory Animals, University Medicine Greifswald, 17489 Greifswald, Germany; 4Boehringer Ingelheim Pharma GmbH & Co. KG, 88400 Biberach, Germany; 5Department of General Pharmacology, University Medicine Greifswald, 17487 Greifswald, Germany

**Keywords:** pharmacobezoars, MRI study, viscosity, spray-dried amorphous solid dispersions, preclinical testing, rodent stomach

## Abstract

Spray-dried amorphous solid dispersions of new chemical entities and pH-dependent soluble polymer hydroxypropyl methylcellulose acetate succinate (HPMC-AS) were found to form solid agglomerates in the gastrointestinal tract of rodents after oral administration. These agglomerates, referring to descriptions of intra-gastrointestinal aggregated oral dosage forms termed pharmacobezoars, represent a potential risk for animal welfare. Previously, we introduced an in vitro model to assess the agglomeration potential of amorphous solid dispersions from suspensions and how it can be reduced. In this work, we investigated if the in vitro effective approach of viscosity enhancement of the vehicle used to prepare suspensions of amorphous solid dispersions could reduce the pharmacobezoar formation potential following repeated daily oral dosing to rats as well. The dose level of 2400 mg/kg/day used in the main study was determined in a dose finding study carried out in advance. In the dose finding study, MRI investigations were carried out at short time intervals to gain insights into the process of pharmacobezoar formation. Whereas MRI investigations underlined the importance of the forestomach for the formation of pharmacobezoars, viscosity enhancement of the vehicle reduced the incidence of pharmacobezoars, delayed the onset of pharmacobezoar formation and reduced the overall mass of pharmacobezoars found at necropsy.

## 1. Introduction

Predicting the safety of a new chemical entity (NCE) prior to first-in-human administration is a main aspect of preclinical testing [1]. Despite new methods of toxicological testing arising from rapid advancements of in vitro models, including three-dimensional cell cultures, microphysiological systems and computational modelling, the complexity of a living organism still prevents the full replacement of animal studies [2]. In the case of toxicity studies, adequate preclinical safety testing in one rodent and one non-rodent species is a mandatory part of preclinical safety testing in order to maximize insight on the NCEs toxicological profile [3]. In a series of rodent repeat-dose toxicity studies and studies on embryofetal development, the formation of pharmacobezoars in the gastrointestinal tract was observed. In these studies, a spray-dried amorphous solid dispersion (SD-ASD) of the NCE BI 1026706 and hydroxypropyl methylcellulose acetate succinate (HPMC-AS) was administered as a suspension in an acidified aqueous vehicle [4]. Apart from sporadic findings in mice, rats were predominantly affected by pharmacobezoar formation. Observed incidences of pharmacobezoars positively correlated with dose and study duration. The lowest dose leading to pharmacobezoar formation was 571 mg SD-ASD administered once daily for 13 weeks. Pharmacobezoars were not observed following administration of pure spray-dried HPMC-AS in doses equal to the share of polymer on the SD-ASD dose administered in the high dose groups [4]. As pharmacobezoars from SD-ASDs were also absent in other test species such as rabbits or cynomolgus monkeys, we hypothesized that the anatomy of the rat stomach may be an important factor. In particular, the nonglandular section of the stomach, physiologically acting as a food storage by accommodating bulks of food, is a unique compartment of the rodent stomach that is predestined as an initial location of pharmacobezoar formation due to the absence of peristaltic movements [5,6,7]. In order to prevent SD-ASD from pharmacobezoar formation, an option would probably be the exchange of the polymer of SD-ASDs to a non-pH dependent soluble polymer so that SD-ASDs dissolve at the latest during gastric residence time following oral administration. However, in addition to the outstanding potential of HPMC-AS to maintain supersaturation and inhibit recrystallization [8], it is particularly the pH-dependent solubility that makes HPMC-AS a frequently chosen polymer for preclinical formulation of poorly soluble NCEs. Contrary to pH-independent soluble polymers, HPMC-AS can be utilized to prepare suspensions of SD-ASDs in an acidic vehicle. These HPMC-AS-based SD-ASD particles from orally administered suspensions would not dissolve prior to the rise of the environmental pH following gastric emptying into the duodenum. With dissolution to the supersaturated solutions right at the place of absorption, optimal conditions for high bioavailability can be achieved. In studies in which pharmacobezoars were observed, no clinical symptoms occurred despite their size of up to more than 1 cm in diameter when the pharmacobezoars were localized exclusively in the stomach at necropsy. However, their potential to impair animal wellbeing was observed in cases when pharmacobezoars were expelled to the small intestines [4]. We could demonstrate in vitro that an increase in the viscosity of the vehicle used to suspend the SD-ASDs effectively reduced the agglomeration potential of two HPMC-AS-based SD-ASD formulations [9].

By the in vivo study reported in this work, we intended to evaluate the effect of viscosity enhancement of the vehicle on the extent of pharmacobezoar formation. Furthermore, the time course of the pharmacobezoar formation process was investigated by repetitive magnetic resonance imaging (MRI).

## 2. Materials and Methods

### 2.1. Tested SD-ASD

Administered SD-ASD, provided by Boehringer Ingelheim Pharma GmbH & Co. KG (Biberach, Germany), was the same formulation that led to increased incidences of pharmacobezoars in nonclinical safety testing in rodents [4]. It consisted of 70% BI 1026706 incorporated in a HPMC-AS (Shin-Etsu AQOAT^®^ AS-LG, Shin-Etsu Chemical Co., Ltd., Tokyo, Japan) polymer matrix and was spray-dried from the methanolic solution. Physicochemical properties of the crystalline NCE and the administered spray-dried formulation were described previously [9].

### 2.2. Preparation of Suspensions

The preparation of the suspensions followed the same procedure as the preparation of the suspensions for in vitro tests [9]. Briefly, the required mass of SD-ASD was weighed into a glass beaker and wetted by a small amount of the respective aqueous vehicle before the remaining amount of aqueous vehicle was added under constant stirring. In the dose finding study and for the positive control group (Group A) of the main study, 0.01 N HCl served as vehicle to suspend the formulation. The high-viscous vehicle used to prepare suspensions administered to animals of Group B in the main study was prepared by the addition of 1% (*w*/*w*) hydroxyethylcellulose (Natrosol^®^ 250HX—Ashland, Wilmington, NC, USA) to the vehicle, that had been allowed to swell for 24 h prior to use. The high-viscous vehicle was found to obtain the same shear thinning behavior that is described in the manufacturer brochure for solutions of 1% Natrosol^®^ 250HX in water with a viscosity of about 10 Pa∙s at a shear rate of 0.01/s to 0.1 Pa∙s at a shear rate of 1000/s [10].

### 2.3. Animal Study Protocol

The animal studies were approved by local government authorities (Landesamt für Landwirtschaft, Lebensmittelsicherheit und Fischerei Mecklenburg-Vorpommern application no. 7221.3-1-026/21). Seven-week-old HanWistar rats were obtained from Janvier Labs (Saint-Berthevin Cedex, Le Genest-Saint-Isle, France), randomly allocated to study groups on arrival and allowed to accustom for about one week. The animals were kept in groups of two or three of the same sex in conventional EU type IV cages under specific pathogen-free conditions in animal climate cabinets under standardized conditions (Table 1). For enrichment, wood chips and paper rolls were provided. Food (Standard pellet food, ssniff Spezialdiäten GmbH, Soest, Germany) and drinking water (pH 2.5) were available ad libitum. Each animal was assigned a number and identified uniquely by ear punching.

As the occurrence of pharmacobezoars has shown to be dose- and time-dependent but highly variable in different nonclinical safety studies [4], we conducted a 24-day dose finding study in advance of the main study. We planned to test the pharmacobezoar formation potential following once daily dosing of 1400, 1900 and 2400 mg/kg SD-ASD, suspended in 10 mL/kg 0.01 N hydrochloric acid (HCl) as vehicle with 18 animals distributed to three groups (Table 2). The aim was to find a dose that entails a high incidence of pharmacobezoars without causing overt impairments to the animal’s general condition. Three males and three females per group were included to investigate the time until first pharmacobezoars could be observed by MRI, as well as determining the mass of pharmacobezoars in the stomach at necropsy. As the dose level for the main study could be determined based on the findings of DF 1 and DF 2, the testing of DF 3 was waived.

During the last three days of the acclimatization period, animals were trained on an insertion of an oral feeding cannula (Schlundsonde FTP-15-64, Medical Industrie GmbH, Mettmann, Germany) and bolus administration of 10 mL/kg non-viscosity-enhanced vehicle once daily.

Pre-dose MRI measurement on day 0 was carried out following oral administration of 10 mL/kg non-viscosity-enhanced vehicle labeled with 1% (*w*/*w*) Gadovist, a gadolinium-based MRI contrast agent. Subsequently, MRI measurements were carried out in a short time interval of 4 days (Figure 1) to track the process of pharmacobezoar formation. On days of MRI, suspensions of SD-ASD were labeled with 1% of Gadovist 1.0 mmol/mL (Bayer AG, Germany) as well.

Imaging was consistently conducted in the morning to keep the time of day of this intervention similar for all animals. As MRI measurements took approximately 30 min per animal, this required distribution of the animals of the dose finding study to three runs. These runs with six rats each were completed one after the other. According to the higher number of animals involved, the main study was separated into 4 runs of 24 days each. For all MRI measurements, animals were anaesthetized by isoflurane (1.5–2.0% isoflurane in oxygen). Bepanthen^®^ ophthalmic ointment (Bayer Vital GmbH, Leverkusen, Germany) was applied for eyecare when animals were placed in the MRI scanner. Subsequent to MRI measurement, the animals were allowed to recover from anesthesia under red light and remained under observation for at least 30 min. Based on the results of the dose finding study, the main study was conducted comparing the incidence of pharmacobezoars in Group A (non-viscosity-enhanced vehicle) vs. Group B (high-viscous vehicle) with 14 female rats per group. The reasons for the exclusion of male rats and for extending the MRI measurement interval to 8 days in the main study (Figure 2) are discussed in Section 4. In every run, animals of Group A and Group B were tested in a balanced ratio.

Following terminal MRI measurement on day 24, all animals were killed by cervical dislocation in anesthesia directly following the last MRI. The abdomen was opened, and the stomach, duodenum and jejunum were removed. To extract pharmacobezoars, the jejunum, duodenum and stomach along the greater curvature were opened. All SD-ASD agglomerates were extracted from the stomach and small intestines, rinsed using acidified water and measured in size. To determine the mass of the extracted pharmacobezoars, they were dried at 30 °C until 24 h loss on drying was below 1%.

### 2.4. Magnetic Resonance Imaging (MRI)

MRI investigations were performed under veterinary supervision using a 7.1 Tesla (T) ClinScan 7030 small animal magnetic resonance imager interfaced with a Syngo VB15 console (Bruker Biospin GmbH, Ettlingen, Germany), providing a gradient system operating with a maximum gradient amplitude of 290 mT/m. Anesthetized rats were placed in a prone position on a cradle with the nose in an air stream of isoflurane. A breath sensor was placed below the thorax. A ^1^H transmit-receive circularly polarized volume coil was placed around the central abdomen for signal detection. Transversal images were acquired following the gradient echo localizer by breath-triggered T1-weighted spin echo sequence. The resulting images showed the rat stomach as visualized schematically in Figure 3. However, as the limiting ridge narrows the oval shaped rodent stomach, peripheral transversal layers of the stomach were depicted as two separate areas of enhanced signal intensity. Subsequent to the transversal scan, a three-dimensional (3D) gradient-echo volumetric interpolated breath-hold examination (VIBE) was executed for magnetic resonance imaging of the rat stomach before the rats were allowed to wake up from anesthesia. The VIBE scans were considered when the transversal scans were not sufficiently conclusive. Breath triggering was conducted using an MRI-compatible Small Animal Monitoring & Gating System (SA Instruments INC., Stony Brook, NY, USA). MRI parameters and procedure were identical in the dose finding study and main study (Table 3).

### 2.5. Image Analysis

Images were evaluated using Horos^TM^ (Version 3.3.6, Horos Project). They were assessed for the presence of gastric pharmacobezoars by three trained, independent observers. The diameter of an object had to reach a minimum size of 4 mm when encircled as shown in Figure 4, to be considered a pharmacobezoar. On day 24, the position of pharmacobezoars found in the stomach during necropsy subsequent to MRI measurements could be directly related to the acquired images.

### 2.6. Evaluation Criteria and Statistics

Based on the significant reduction of SD-ASD agglomeration potential by viscosity increasement of the vehicle in vitro [9], a priori sample size calculation for the main study was carried out estimating an effect size (Cohens d) of 1, alpha error 0.05 and power set at 0.8:$$\mathrm{Effect}\;\mathrm{size}=\frac{X_{A}-X_{B}}{\sqrt{\frac{\left(S_{A}^{2}+S_{B}^{2}\right)}{2}}}$$

X: mean values of the pharmacobezoar weight in study groups.S: Standard deviation of the pharmacobezoar weight in study group.A: Group A/B: Group B.

The calculations using G*Power (version 3.1.9.3) resulted in a group size of 14 animals per group of the main study to be sufficient to observe the statistically significant differences of pharmacobezoar formation. These statistical assumptions required a high incidence in the positive control group (Group A). The dose finding study was conducted to determine an appropriate dose level. Statistical evaluations of the differences in pharmacobezoar weight between Group A and Group B of the main study were carried out applying an unpaired *t*-test using GraphPad Prism 5.01 and were considered statistically significant in the case of *p* < 0.05.

## 3. Results

### 3.1. Dose Finding Study

The dose finding study was started with four female and two male rats in the first run, which were equally assigned to DF 1 (2400 mg/kg/day) and DF 2 (1900 mg/kg/day). All six of these animals developed pharmacobezoars in the stomach, whereas no pharmacobezoars in the small intestines were found. As no deterioration of animal wellbeing was observed, we decided to complete DF1 and DF2 in the second run instead of assigning half of these animals to DF 3 (1400 mg/kg/day) as initially planned.

With the exception of one male dosed with 1900 mg/kg/day, pharmacobezoars were present during necropsy of all rats of the second run as well. Characteristically, a major pharmacobezoar extended from the dome of the forestomach in the direction of the limiting ridge and dilated the forestomach (Figure 5). Pharmacobezoars were, apart from an occasionally observed small hole at the dorsal side of the main pharmacobezoar, solid at contact surfaces to the gastric mucosa. Contrary, luminal surfaces were not entirely compacted and showed a more irregular texture. Remarkably, pharmacobezoars did regularly not include any food particles. In a few cases, some hairs were enclosed. Adjacent to the solid pharmacobezoars in the forestomach, the food pulp was found to be encased by small, non-rigid agglomerates of ASD (Figure 5—3) that were predominantly too small or not solid enough to be classified as pharmacobezoars. In the glandular part, the density of small agglomerates was lower when compared to the nonglandular part and the observed agglomerates were predominantly incorporated in the food pulp. In the dose finding study, no pharmacobezoars were found in the small intestines.

The total mass of the solid pharmacobezoars recovered from the stomach during necropsy ranged from 184 mg to 1318 mg per animal (Figure 6). The both highest weights were observed in two males of DF 1 that had also received the highest total dose of SD-ASD according to their high body weight. Interestingly, the major pharmacobezoar was found to be larger in females (mean mass 204 mg) than in males (mean mass 121 mg), were the major pharmacobezoar was frequently accompanied by several pharmacobezoars of only slightly smaller size (Figure 7). In numbers, the major pharmacobezoars accounted for 60% of the total mass of pharmacobezoars in females on average, whereas the major pharmacobezoar in males accounted only for 25% of the total mass of pharmacobezoars.

Like the mass of pharmacobezoars, the study day when a pharmacobezoar was first visible in the stomach via MRI was highly variable. First pharmacobezoars were already detectable in one female of DF 1 and DF 2 at the first post-dose MRI measurement following 4 days of dosing. At the third MRI measurement on day 8, pharmacobezoars were present in 60% of animals, whereby the onset of pharmacobezoar formation tended to be earlier in DF 1 (Table 4). In DF 2, no pharmacobezoar formation was observed in one male.

Besides enabling a comparison of the temporal occurrence of pharmacobezoars, MRI examinations also allowed a deeper insight into the process of pharmacobezoar formation. In all rats, the initial stages of pharmacobezoar formation took place in the dorsal forestomach, where also the major pharmacobezoars were observed at necropsy. Two-dimensionally, shapes of pharmacobezoars in the transversal plane in MRI images ranged from oval to crescent, depending on the plane of the stomach in the respective transversal image. Three-dimensionally, pharmacobezoars were chalice shaped and partially enveloped the food pulp in the forestomach, which was permeated with a contrast agent, as shown in the middle image in Figure 8.

In subsequent stages of the pharmacobezoar formation process, pharmacobezoars were mostly found to have varying shapes. Typically, pharmacobezoars showed at some point break edges, which were clearly visible when a contrast agent was present in the interstitial space (Figure 8—right). If multiple pharmacobezoars were present, their location in consecutive MRI scans often changed. However, the major pharmacobezoar and other, especially larger accompanying pharmacobezoars, predominantly remained in the forestomach until necropsy.

### 3.2. Main Study

Based on the observations of the dose finding study, we chose the dose tested in DF 1 (2400 mg/kg/day SD-ASD suspended in 10 mL/kg of the respective vehicle) to investigate the effect of increased vehicle viscosity on pharmacobezoar formation. Due to the unexpected losses of three males in anesthesia during the pre-dose MRI measurement of the dose finding study and the trend of earlier onset of pharmacobezoar formation in females observed via MRI, we excluded male rats from the main study. Furthermore, we decided to double the interval between MRI measurements to reduce associated stress for the animals in the main study and compared the incidence of pharmacobezoars on days 8, 16 and 24.

Extensive pharmacobezoar formation in Group A, receiving 0.01 N HCl as non-viscosity-enhanced vehicle, was reflected by the onset of pharmacobezoar formation until first post-dose MRI measurement on day 8 for 70% of the 14 animals (Figure 9). Within 16 days of dosing, the incidence of pharmacobezoars reached 100%. The incidence of pharmacobezoars in Group B, receiving the same dose SD-ASD suspended in high-viscous vehicle, was significantly lower at MRI measurements with 4 animals affected on day 8 and 6 animals on day 16. On the last study day, MRI measurements revealed that pharmacobezoars were still absent in three animals of Group B, which was confirmed at necropsy. Thus, Group B comprises only 11 values in Figure 10. Comparing the mean mass of pharmacobezoars observed in Group A and Group B, we observed a significant reduction of 69%.

## 4. Discussion

The selected doses and study duration for the dose finding study were based on the occurrence of pharmacobezoars in previous nonclinical safety studies with the same SD-ASD formulation. Pharmacological effects of the NCE (a bradykinin 1 receptor antagonist) were not expected, as it already showed a low affinity and no test item-related toxicologically relevant effects in rats [11]. As SD-ASD doses of 2143 mg/kg/day and 2858 mg/kg/day (1428 mg/kg/day twice daily) resulted in the occurrence of pharmacobezoars in all animals within 7, respectively 5 days [4], we expected the pharmacobezoar formation process to start shortly after the beginning of the study. Therefore, a short time interval of 4 days between pre-dose and second MRI was selected to capture the initial phase of pharmacobezoar formation in the dose finding study. Defining a suitable study duration, we knew that long study durations increase the probability of high pharmacobezoar incidences, even though this also raises the risk of pharmacobezoar-related impairments of animal welfare. Furthermore, we wanted to keep the time interval between MRI investigations constant. We decided on a study duration of 24 days, in which a total of 7 MRI measurements were conducted to enable continuous insights into shape and position of pharmacobezoars during their formation process by noninvasive visualization of the intragastric lumen in addition to the final snapshots obtained from necropsies. To enhance contrast between solid pharmacobezoars and other gastric contents, we labeled the suspensions of SD-ASDs administered at days of MRI measurements with a gadolinium-containing contrast agent. As free gadolinium is highly toxic and the stability of gadolinium complexes in the harsh gastrointestinal environment is stressed anyway, we chose to apply a macrocyclic contrast agent rather than the linear chelate complex gadopentetate dimeglumine (Magnevist^®^) that was previously used to enhance signal intensity in T1-weighted MRI measurements of the stomach of rodents [12]. Specifically, we labeled the vehicle of the SD-ASD suspensions administered prior to MRI measurements with 1% (*w*/*w*) of the macrocyclic contrast agent Gadobutrol (Gadovist^®^) obtaining a high kinetic stability [13]. To prove the MRI concept in advance of the dose finding study, T1-weighted MRI scans with SD-ASD agglomerates from in vitro experiments embedded in a mixture of food pulp and SD-ASD suspension labeled with contrast agent were conducted. In vitro agglomerates, depicted as dark areas in these acquired images, were well distinguishable from the contrast agent labeled food pulp. 

First evaluation of the MRI images from the study started with the MRI scans generated directly prior to necropsy to confirm that the position, shape and size of the pharmacobezoars identified via MRI were identical to what was observed at necropsy. Compact, solid areas of pharmacobezoars could be clearly demarcated as dark areas in the acquired images. Typically, this included areas where the surface of the pharmacobezoars was rigid due to permanent contact with the stomach wall or at fracture points of the pharmacobezoars. Especially when the food pulp showed heterogenic signal intensity, demarcation of loosely agglomerated areas of the pharmacobezoars or of small, friable particles of agglomerated SD-ASD was not possible. Therefore, SD-ASD agglomerates were only classified as a pharmacobezoar when they had a diameter of 4 mm in the MRI scans (Figure 4) or ex vivo. Based on these limitations, a determination of the volume of the pharmacobezoars based on MRI was not performed.

Besides the study duration, the SD-ASD dose is the second parameter known to have major influence on pharmacobezoar formation. To assure that high incidences of pharmacobezoars occurred within the study duration chosen, the main study was preceded by a dose finding study including three groups of three female and three male rats each at dose levels of 2400, 1900 and 1400 mg/kg/day. The higher dose was tested first to maximize the likelihood of pharmacobezoar formation. The extensive pharmacobezoar formation without impairment of animal wellbeing observed in the first run of the dose finding study encouraged to complete DF 1 and DF 2 before testing DF 3. As all rats of DF 1 developed a pharmacobezoar without impaired general condition, 2400 mg/kg/day was chosen as the dose to be tested in the main study. Like the decision to start the investigation with DF 1 and DF 2, the choice of short MRI intervals also proved to be beneficial to detect differences in the onset of pharmacobezoar formation between both groups as well as between females and males. Whereas the onset of pharmacobezoar formation in the dose finding study was overall highly variable, the females of DF 1 stood out with a steady onset of pharmacobezoar formation on day 4 or day 8.

The initial location of pharmacobezoar formation was always the forestomach. As a certain amount of air is present in the forestomach, most pharmacobezoars obtained a hole at the point which, in physiological position, corresponds to the highest point in the stomach (Figure 5—right). With respect to rodent anatomy and physiology, the localization of the formation process in the forestomach is conclusive: Besides a constant tonus, no contractile motility occurs in this compartment that is known to basically serve as a food storage compartment. The typically observed characteristic chalice shape of the major pharmacobezoar enveloping the food pulp (Figure 8—middle) might be the consequence of an always similarly proceeding process of pharmacobezoar formation. While the gastric-emptying rate of food pulp from the forestomach of rodents is constant over hours, liquids are emptied much faster [14,15,16,17,18]. It is known from other species that liquids can pass the food pulp along the stomach wall without significantly mixing with the food pulp [19,20,21]. Transferring this emptying process of liquids to the situation after dosing, SD-ASD suspensions probably distributed between the mucosa and the food pulp in the forestomach. The increase in intragastric pressure resulting from the volume increasement can be compensated by distension of the elastic stomach wall, which however still provides a constant tonus on the luminal content. Subsequently ongoing agglomeration processes and parallel onflow of the suspensions liquid phase leaves behind a layer of agglomerated SD-ASD particles enveloping the food pulp, which might be further compacted by the constant tonus. From this point, the repetition of this process with each dose of SD-ASD suspension would contribute to further increases in the pharmacobezoar size. The localization of loosely agglomerated SD-ASD particles that presumably remained from the last administration about 45 min before necropsy, surrounding the outer side of the pharmacobezoars and food pulp in the forestomach, promotes the formerly presented explanation.

At later stages of the formation process, MRI scans revealed that, particularly in males, instead of a steady increase in wall thickness of the chalice, pharmacobezoars varied their spatial shape as parts might have broken off and rearranged. A reason for the formation of comparatively small major pharmacobezoars accompanied by multiple medium-sized pharmacobezoars observed in males (Figure 7) could be their higher food consumption and their higher body mass [22]. Whereas the latter might favor pharmacobezoar formation due to the higher total dose of SD-ASD administered, increased mass transfer in the stomach due to higher food consumption might entail the fragmentation of small pharmacobezoars, thus favoring mixing with the food pulp and consequent emptying together with non and loosely agglomerated SD-ASD from the stomach. Nevertheless, the bigger pharmacobezoars were predominantly found to be retained in the forestomach.

In the dose finding study, three male rats did not awake from inhalative isoflurane narcosis following pre-dose MRI measurement due to breath depression. These rats had not received any SD-ASD at this timepoint and at necropsy no obvious reasons for premature deaths could be determined. As indications for the accumulation of the lipophilic anesthetic isoflurane in subcutaneous fat in mice are reported [23] and male rats obtain a higher volume of distribution for lipophilic drugs corresponding to the higher body fat content [24], a correlation between the incidents and increased accumulation of isoflurane might be conclusive. To counter further animal losses and simultaneously improve the consistency of pharmacobezoar formation within the positive control group of the main study in order to maximize the assessability of the influence of the viscosity of the vehicle, we decided to exclude males from the main study.

As insights on the process and initial location of pharmacobezoar formation were already evident from tightly scheduled MRI measurements in the dose finding study, their frequency in the main study was reduced to an interval of 8 days. Thereby, we aimed to reduce stress for the animals associated with the measurement procedure [25] along with possible influences of anesthesia on gastrointestinal motility [26], while keeping the ability of comparing the time-dependent occurrence of pharmacobezoars in both groups. Another change in the study protocol from the dose finding study to the main study was the extension of the time period between administration and MRI measurements from 15 min to 30 min for animals receiving the high-viscous vehicle. We observed in the first pre-dose MRI scans of Group B that diffusion of the contrast agent from the high-viscous vehicle into the food pulp was markedly slowed down compared to the non-viscosity-enhanced vehicle. Besides the affection of physical parameters relevant for the agglomeration of suspended particles, such as particle interactions and mobility, it should be mentioned that the viscosity of administered suspensions might affect physiological parameters that are probably relevant for pharmacobezoar formation as well. Increased viscosity is known to reduce the rate of gastric emptying, to enhance the gastric residence time and to slow down fluid motions in the stomach [5,14,27]. Projecting this on the situation following the dosing of SD-ASD suspended in high-viscous vehicle, the previously discussed process of rapid onflow of the liquid phase from standard suspensions, leaving behind the dispersed solid phase in the lumen between mucosa and food pulp in the forestomach, would be impeded. All these factors that the use of the high-viscous vehicle entails resulted in a reduction of 69% of the mean mass of the pharmacobezoars and the absence of pharmacobezoars in three females at necropsy following 24 days of dosing the viscosity-enhanced suspension. The time-dependent formation process was furthermore found to be delayed. In a two-week study, for instance, pharmacobezoar formation would have been prevented in 50% of the animals by application of the high-viscous vehicle. 

However, the results show that increasing the viscosity of the vehicle cannot completely prevent pharmacobezoar formation. Furthermore, it should be mentioned that the design of our study precluded a conclusion regarding the degradation of pharmacobezoars after the cessation of treatment.

## 5. Conclusions

The aim of this study was to gain insights in the process of pharmacobezoar formation as well as to validate the approach of increasing the viscosity of the vehicle used to prepare suspensions of SD-ASDs in order to impede pharmacobezoar formation in vivo. As the latter required a sufficient incidence of pharmacobezoars to evaluate the effect, we conducted a dose finding study prior to the comparative main study. Using non-invasive MRI technique, we were able to gain insights into the process of pharmacobezoar formation during the 24-day study period that could not be obtained from terminal examinations of pharmacobezoars at necropsy. In all cases, the initial location of pharmacobezoar formation was the forestomach. The acidic environment without relevant motility is considered to provide conditions that promote the agglomeration of HPMC-AS-based SD-ASD particles to solid pharmacobezoars. Increasing the viscosity of the vehicle used to prepare the suspensions of SD-ASDs by the addition of 1% HEC effectively reduced the incidence of pharmacobezoars, their mean mass found at necropsy and prolonged the time span until the pharmacobezoars could be detected via MRI.

Altogether, the results show that the approach of vehicle viscosity enhancement can be effectively utilized in vivo to minimize risks of animal welfare impairments due to the formation of pharmacobezoars from SD-ASD that revealed a high in vitro agglomeration potential.

## Figures and Tables

**Figure 1 pharmaceutics-15-00887-f001:**
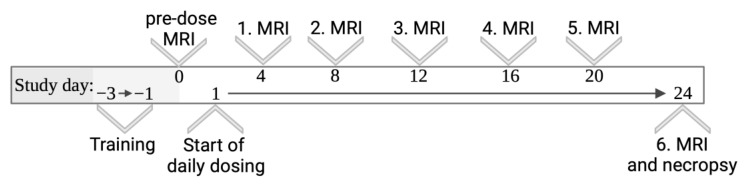
Study design—dose finding study.

**Figure 2 pharmaceutics-15-00887-f002:**
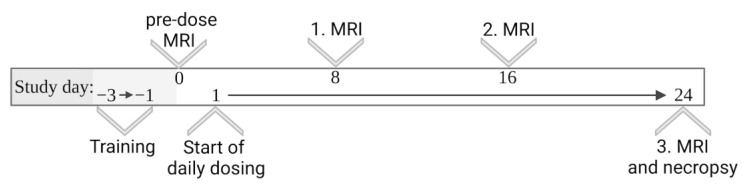
Study design—main study.

**Figure 3 pharmaceutics-15-00887-f003:**
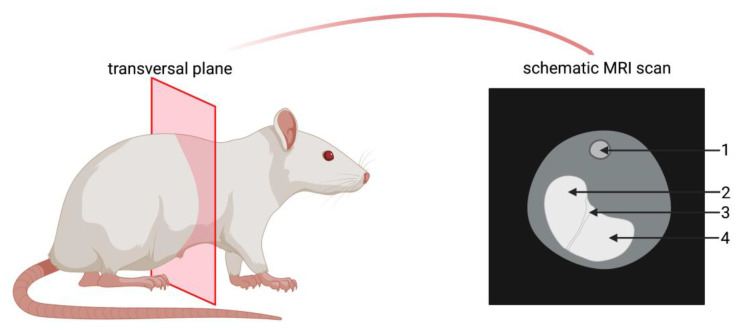
Visualization of transversal plane (**left**) and schematic MRI scan of the stomach in this plane (**right**). 1—spine; 2—forestomach; 3—limiting ridge; 4—glandular stomach.

**Figure 4 pharmaceutics-15-00887-f004:**
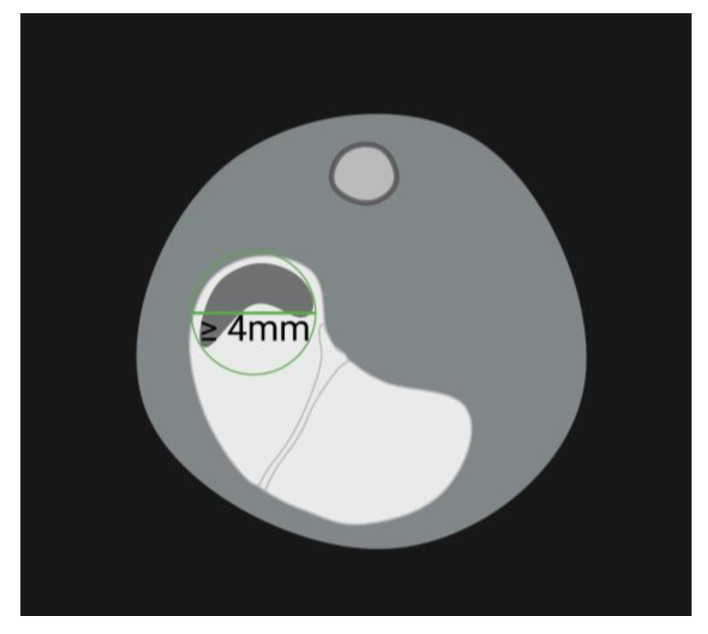
Schematic evaluation of the maximum diameter of a pharmacobezoar in forestomach. ROI: green circle; green horizontal line = maximum diameter.

**Figure 5 pharmaceutics-15-00887-f005:**
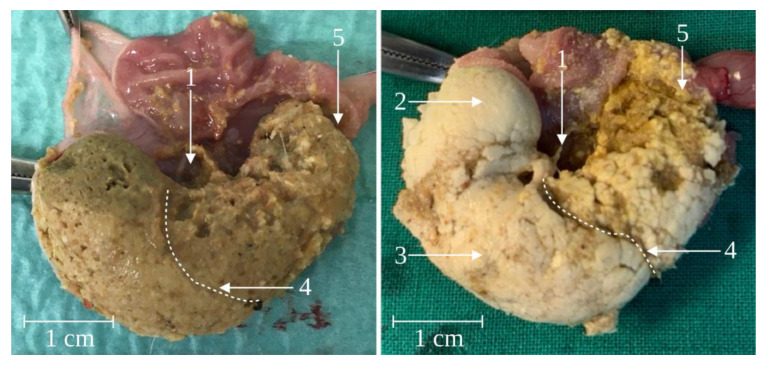
Ex vivo opened rat stomach of a female rat without pharmacobezoar (**left**) and of a rat from group DF 1 with pharmacobezoar (**right**) (1—cardia, 2—major solid pharmacobezoar with small hole at the upper left side, 3—non-rigid SD-ASD particles, 4—limiting ridge, 5—pylorus).

**Figure 6 pharmaceutics-15-00887-f006:**
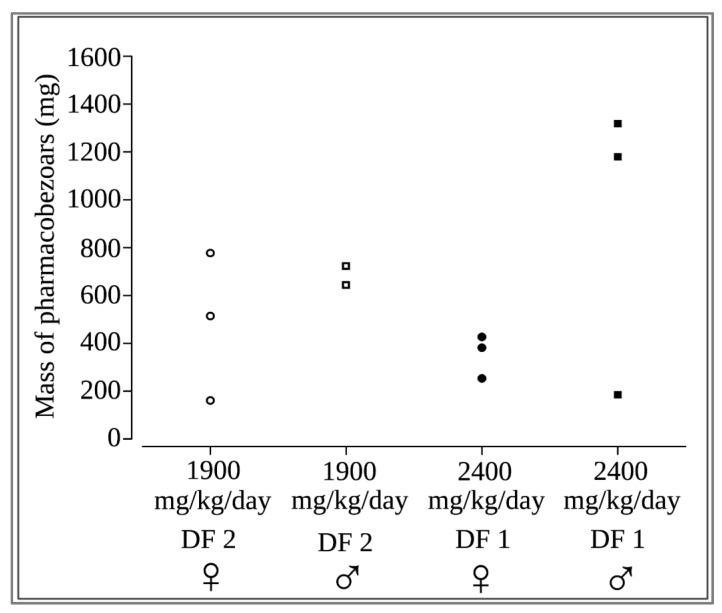
Total mass of all pharmacobezoars observed per rat in the dose finding study (♀: female; ♂: male).

**Figure 7 pharmaceutics-15-00887-f007:**
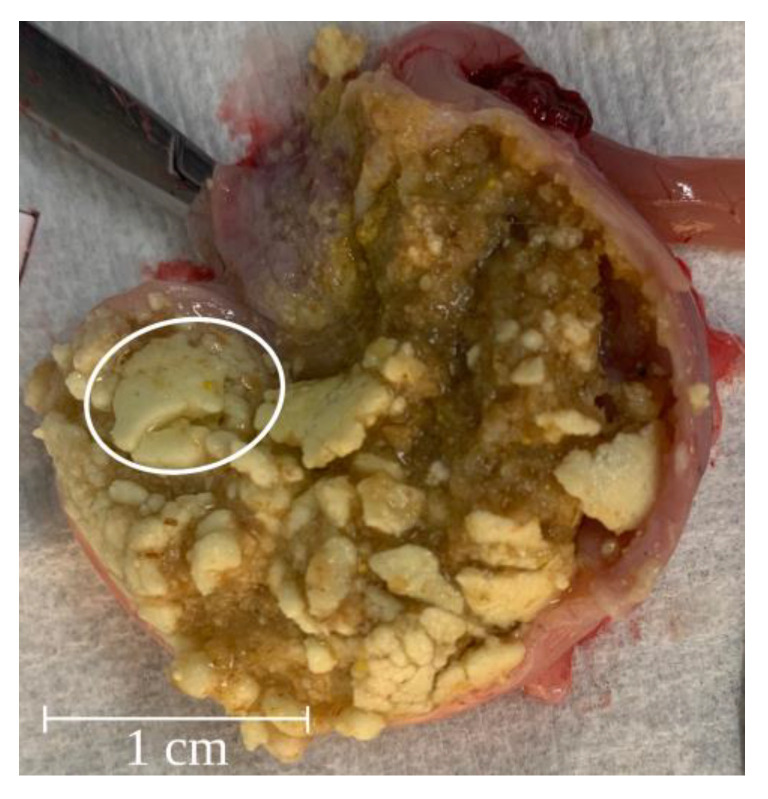
At scheduled necropsy opened rat stomach of a male rat of DF 1 with multiple small to medium-sized pharmacobezoars accompanying the encircled major pharmacobezoar.

**Figure 8 pharmaceutics-15-00887-f008:**
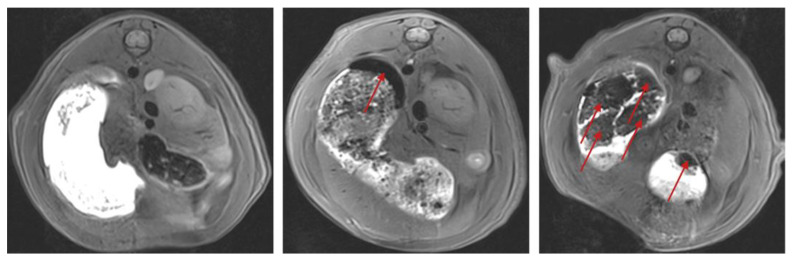
Pre-dose MRI image and MRI images of pharmacobezoars marked by red arrows at day 8 and day 16 in the stomach of a female rat after dosing of 2400 mg/kg/day SD-ASD in non-viscosity-enhanced vehicle (T1-weighted, transversal). Pre-dose MRI (**left**); onset of pharmacobezoar formation on day 8 with typical crescent shape of pharmacobezoar in the forestomach (**middle**) and pharmacobezoars observed on day 16 in the forestomach obtaining broken edges plus a smaller pharmacobezoar in the glandular stomach (**right**).

**Figure 9 pharmaceutics-15-00887-f009:**
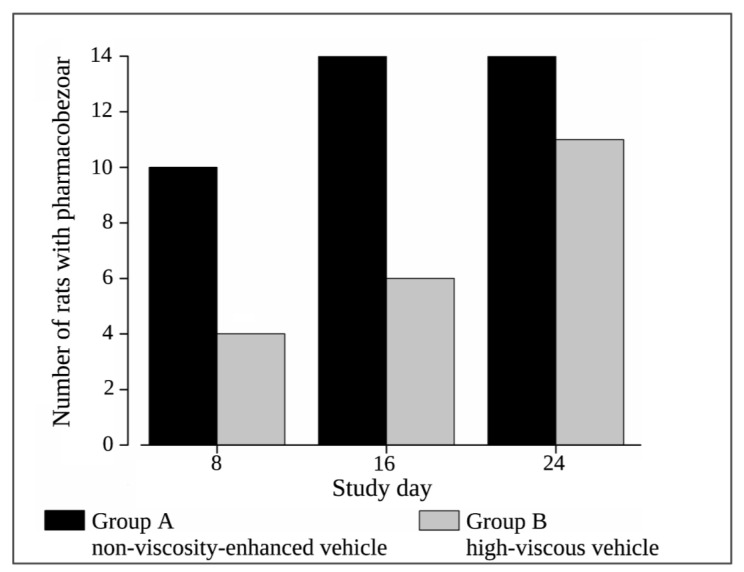
Incidence of pharmacobezoars observed at MRI measurements on day 8, 16 and 24 of the main study in Group A (non-viscosity-enhanced vehicle) and Group B (high-viscous vehicle); both groups *n* = 14.

**Figure 10 pharmaceutics-15-00887-f010:**
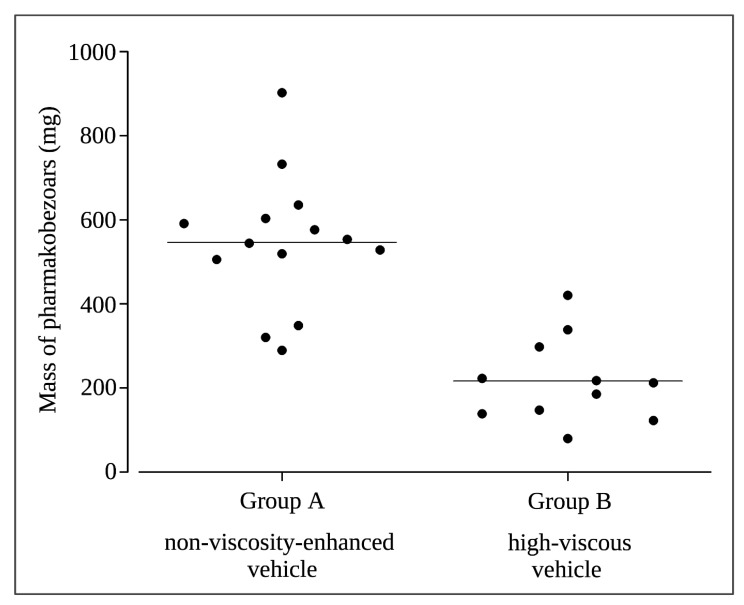
Mass of pharmacobezoars at necropsy after 24 days dosing of 2400 mg/kg/day BI 1026706 SD-ASD suspended in non-viscosity-enhanced vehicle (Group A) and high-viscous vehicle (Group B); mean values are indicated by horizontal line.

**Table 1 pharmaceutics-15-00887-t001:** Animal housing conditions.

Temperature (°C)	20 ± 2
Relative Humidity (%)	45–75
Light/dark cycle (h)	12/12
Air changes per hour	max. 12

**Table 2 pharmaceutics-15-00887-t002:** Overview of dose finding study and main study.

Study	Group	Dose SD-ASD (mg/kg/d)	Vehicle	Number of Animals
Dose finding study	DF 1	2400	0.01 N HCl	6
	DF 2	1900	0.01 N HCl	6
	*DF 3 **	*1400*	*0.01 N HCl*	*6*
Main study	A	2400	0.01 N HCl	14
	B	2400	0.01 N HCl + 1% HEC (*w*/*w*)	14

* Testing of DF 3 was waived.

**Table 3 pharmaceutics-15-00887-t003:** MRI parameter for all evaluations.

Parameter	T1-Weighted Spin Echo	T1-Weighted Flash 3D Vibe Sequence
Repetition time	827 ms	15 ms
Echo time	11 ms	1.42 ms
Slice thickness	1.5 mm	0.5 mm
Interslice gap	1.65 mm	-
Flipangle	90°	25°
Matrix size	512 × 512	512 × 512
Field of view	71 × 71	71 × 71

**Table 4 pharmaceutics-15-00887-t004:** First day a pharmacobezoar was visible by MRI in the stomach of rats in the dose finding study. Dose of spray-dried amorphous solid dispersion given in brackets.

Group	DF 1 (2400 mg/kg/day)						DF 2 (1900 mg/kg/day)					
Sex	♂			♀			♂			♀		
Study day	8	8	24	4	8	8	-	8	12	4	8	20

## Data Availability

Not applicable.

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
