# Peer review of "Evaluation of Pharmacobezoar Formation from Suspensions of Spray-Dried Amorphous Solid Dispersions: An MRI Study in Rats"

_pharmaceutics, 2023, doi:10.3390/pharmaceutics15030887_

Round 1

Reviewer 1 Report

·       Can you discuss the in vitro model used to assess the agglomeration potential of amorphous solid dispersions?

·       How did the viscosity enhancement of the vehicle used to prepare suspensions impact the formation of pharmacobezoars in rats after repeated daily oral dosing?

·       Can you discuss the results of the MRI investigations carried out in the dose finding study?

·       How did the MRI investigations contribute to the understanding of the process of pharmacobezoar formation?

·       Can you discuss the importance of the forestomach in the formation of pharmacobezoars?

·       How did viscosity enhancement of the vehicle affect the incidence and onset of pharmacobezoar formation and the overall mass of pharmacobezoars found at necropsy?

·       What are the future implications of your study on the development and administration of new chemical entities?

·       How does this study add to the current understanding of pharmacobezoar formation in rats?

·       What were the key findings of this study and what implications does it have for drug delivery?

Author Response

Answers to comments and suggestions of Reviewer 1

Dear reviewer, we appreciate your efforts in order to improve the manuscript and the valuable comments.

Can you discuss the in vitro model used to assess the agglomeration potential of amorphous solid dispersions?

We stated from line 71 to line 74 that we found an in vitro approach to reduce the agglomeration potential by application of the in vitro model. The scope of the present work is now the in vivo validation of this approach to reduce the formation of pharmacobezoars. For a detailed description of the background of the model development, the mechanical function and discussion including limitations of the model, please see the cited manuscript entitled “An in vitro model to investigate the potential of solid dispersions to form pharmacobezoars.” (reference 9).

How did the viscosity enhancement of the vehicle used to prepare suspensions impact the formation of pharmacobezoars in rats after repeated daily oral dosing?

The viscosity of the vehicle did not change the initial location of pharmacobezoars (line 280). However, it reduced the incidence of pharmacobezoars, effectively delayed the onset of pharmacobezoar formation (Figure 9) and reduced the mass of pharmacobezoars found at necropsy (Figure 10). “All these factors that the use of the high-viscous vehicle entails resulted in a reduction of 69% of the mean mass of the pharmacobezoars and the absence of pharmacobezoars in three females at necropsy following 24 days of dosing of the viscosity-enhanced suspension. The time-dependent formation process was furthermore found to be delayed. In a two week study for instance, pharmacobezoar formation would have been prevented in 50% by application of the high-viscous vehicle. However, the results show that increase of the viscosity of the vehicle cannot completely prevent pharmacobezoar formation. The design of our study precluded a conclusion regarding degradation of pharmacobezoars after cessation of treatment.”(line 458-466)

Can you discuss the results of the MRI investigations carried out in the dose finding study?

We included following discussion section in the manuscript:

“Whereas the onset of pharmacobezoar formation in the dose finding study was overall highly variable, the females of DF 1 stood out with a steady onset of pharmacobezoar formation on day 4 or day 8.

The initial location of pharmacobezoar formation was always the forestomach. As a certain amount of air is present in the forestomach, most pharmacobezoars obtained a hole at the point which, in physiological position, corresponds to the highest point in the stomach (Figure 5). “ line (389-395)

and

“At later stages of the formation process, MRI scans revealed that particularly in males, instead of a steady increase in wall thickness of the chalice, pharmacobezoars varied their spatial shape as parts might have broken off and rearranged. A reason for the formation of comparatively small major pharmacobezoars accompanied by multiple medium-sized pharmacobezoars observed in males (Figure 7) could be their higher food consumption and their higher body mass [21]. Whereas the latter might favor pharmacobezoar formation due to the higher total dose of SD-ASD administered, increased mass transfer in the stomach due to higher food consumption might entail fragmentation of small pharmacobezoars, thus favoring mixing with the food pulp and consequent emptying together with non and loosely agglomerated SD-ASD from the stomach. Nevertheless, the bigger pharmacobezoars were predominantly found to be retained in the forestomach.” (line 416 – 426)

This discussion refers to the results presented from line 274 – 302.

How did the MRI investigations contribute to the understanding of the process of pharmacobezoar formation?

The MRI investigations gave insights in the location and shape of pharmacobezoars during their formation process and thereby underlined that the forestomach plays an important role for pharmacobezoar formation in rodents. Please see also the answer to the following question and the question before, where the corresponding explaining parts of the manuscript are cited that connect insights from MRI investigations to anatomical and physiological parameters that are important for the formation of pharmacobezoars in rodents.

Can you discuss the importance of the forestomach in the formation of pharmacobezoars?

The forestomach as unique anatomical feature of the rodent stomach might be the reason for the extensive pharmacobezoar formation. We stated following in the manuscript: “With respect to rodent anatomy and physiology, the localization of the formation process in the forestomach is conclusive: Besides a constant tonus, no contractile motility occurs in this compartment that is known to basically serve as a food storage compartment. The typically observed characteristic chalice shape of the major pharmacobezoar enveloping the food pulp (Figure 8- middle) might be the consequence of an always similarly proceeding process of pharmacobezoar formation. While the gastric-emptying rate of food pulp from the forestomach of rodents is constant over hours, liquids are emptied much faster [13–17]. It is known from other species, that liquids can pass the food pulp along the stomach wall without significant mixing with the food pulp [18–20]. Transferring this emptying process of liquids to the situation after dosing, SD-ASD suspensions probably distributed around the food pulp and the mucosa in the forestomach. The increase in intragastric pressure resulting from the volume increasement can be compensated by distension of the elastic stomach wall, that however still provides a constant tonus on the luminal content. Subsequently ongoing agglomeration processes and parallel onflow of the suspensions liquid phase leaves behind a layer of agglomerated SD-ASD particles enveloping the food pulp, that might be further compacted by the constant tonus. From this point, repetition of this process with each dose of SD-ASD suspension would contribute to further increase of pharmacobezoar size. The localization of loosely agglomerated SD-ASD particles that presumably remained from the last administration about 45 min before necropsy, surrounding the outer side of the pharmacobezoars and food pulp in the forestomach, promotes the formerly presented explanation.”

How did viscosity enhancement of the vehicle affect the incidence and onset of pharmacobezoar formation and the overall mass of pharmacobezoars found at necropsy?

Both, incidence and onset of pharmacobezoar formation were reduced, respectively delayed. Please see Figure 9 and Figure 10, that illustrate the results. Both figures are complemented by following statement: “Extensive pharmacobezoar formation in Group A, receiving 0.01 N HCl as non‑viscosity‑enhanced vehicle, was reflected by onset of pharmacobezoar formation until first post‑dose MRI measurement on day 8 for 70% of the 14 animals (Figure 9). Within 16 days of dosing the incidence of pharmacobezoars reached 100%. The incidence of pharmacobezoars in Group B, receiving the same dose SD-ASD suspended in the high-viscous vehicle, was significantly lower at MRI measurements with 4 animals affected on day 8 and 6 animals on Day 16. At the last study day, MRI measurements revealed that pharmacobezoars were still absent in three animals of Group B, which was confirmed at necropsy. Thus, Group B comprises only 11 values in Figure 10. Comparing the mean mass of pharmacobezoars observed in Group A and Group B, we observed a significant reduction of 69%.” (line 315-325)

What are the future implications of your study on the development and administration of new chemical entities?

Our aim was to prevent negative consequences of pharmacobezoar formation on the informative value of preclinical studies and animal wellbeing. This study proofs that this can be implemented by application of the in vitro model before administration of SD-ASDs. Therefore, we stated:

„Altogether, the results show that the approach of vehicle viscosity enhancement can be effectively utilized in vivo to minimize risks of animal welfare impairments due to pharmacobezoar formation of SD-ASD that revealed a high in vitro agglomeration potential.“

line 481-484

How does this study add to the current understanding of pharmacobezoar formation in rats?

To the best of our knowledge, no study on the formation of pharmacobezoars has been reported previously. Thus, this study reflects the current understanding of pharmacobezoar formation in rodents and shows that further work is needed in order to not only reduce, but also understand  the reasons for high pharmacobezoar formation potential of SD-ASDs. The key to that question would be other SD-ASDs we don’t know about jet, that have induced pharmacobezoars during preclinical testing. We hope, that by raising attention to this problem, we might induce awareness for this problem.

What were the key findings of this study and what implications does it have for drug delivery?

Pharmacobezoars represent a potential problem regarding animal welfare and the informative value of preclinical studies. With an increasing share of poorly soluble NCEs under development and a correspondingly higher share of SD-ASDs that might be used for adequate preclinical testing, the problem gets even more relevant. Our study should contribute to the Refinement of preclinical studies on rodents.

Reviewer 2 Report

The manuscript “Evaluation of Pharmacobezoar Formation from Suspensions of Spray-dried Amorphous Solid Dispersions:  An MRI Study in Rats” by Gierke, et al. presents new results characterizing formation of in vivo agglomerate in the rat stomach, which could be interesting to readers of Pharmaceutics.  The assessment of pharmacobezoar formation in the imaging studies appear to be generally well done.  In my opinion, information from control experiments, and some additional characterization data of the described vehicles could provide more context and increase the value of the reported results.  Detailed comments are below.

1.      The doses studied are high.  The authors stated their aim to use a “dose that entails a high incidence of pharmacobezoar without causing overt impairment of the animal’s general condition.”  (lines 119-120)  They used 2400 mg/kg/day of ASD for three weeks.  At 70% drug load, this is 1680 mg/kg/day of a drug having an aqueous solubility of 20 microgr/mL.  I wonder if only a very small fraction of this drug would be absorbed.  As this seems a very high dose for this API and other poorly soluble APIs likely to be formulated as ASDs, it would be interesting to know the incidence of the pharmacobezoar at lower API/ASD doses.  Is this phenomenon only a concern at the very high doses shown here (1900+ mg/kg/day ASD), or does it manifest at lower doses?

2.      Related, I could not tell if the authors or others had previously done MRI or other experiments on placebo HPMCAS particles prepared by spray drying or other techniques to look for the pharmacobezoar as a control experiment to determine whether the HPMCAS alone is largely responsible for the agglomerates.  The in vitro work previously reported by some of the present authors in ref. 9 also seems to have used similarly high-loaded ASDs of HPMCAS.  To what extent does the identity of the drug or the drug loading matter in formation of the agglomerates?   The placebo HPMCAS experiment seems an important control to understand the cause of agglomerate formation, and to guide the readers who may want to design formulations for suspension dosing in rats for other drug-containing ASDs with HPMCAS.

3.      The ASDs were dosed in an acidic vehicle.  Is this an important factor in causing the pharmacobezoar?  Many times HPMCAS-based ASDs can be dosed in a higher pH suspension vehicle.  Does this mitigate or completely prevent the formation of pharmacobezoar because the HPMCAS is more soluble at higher pH values?  This seems important to know.

4.      It was found in the previous in vitro study (cf. ref. 9), and in the present in vivo work, that adding HEC to the aqueous vehicle significantly mitigates the agglomerate formation.  The authors attribute this to increased viscosity of the HEC vehicle.  Maybe I missed it in the discussion, but it was not clear to me why a higher viscosity vehicle would mitigate agglomerate formation.  Is the mechanism known or obvious?  Assuming viscosity is an important factor, it would seem useful to know the viscosity of the suspensions used in the current manuscript.  What was the viscosity of the HCl vehicle vs. the HEC-based vehicle?  I assume the viscosity of a suspension having a lot of HPMCAS will depend on the pH.  Would the viscosity likely be significantly higher at higher pH for the suspensions reported here, i.e. 72 mg/mL HPMCAS-L?  Some more quantitative and detailed assessment of the viscosity would be valuable for a reader to understand and potentially use in design of suspension vehicles for dosing.

Author Response

Answers to comments and suggestions of Reviewer 2

Dear reviewer, we appreciate your efforts in order to improve the manuscript and the valuable comments. You pointed out several questions that were also discussed during manuscript preparation. We discussed your comments (italic letters) in the following answers and look forward hearing back from you if there are still points that are not sufficiently explained from your point of view.

The doses studied are high.  The authors stated their aim to use a “dose that entails a high incidence of pharmacobezoar without causing overt impairment of the animal’s general condition.”  (lines 119-120)  They used 2400 mg/kg/day of ASD for three weeks.  At 70% drug load, this is 1680 mg/kg/day of a drug having an aqueous solubility of 20 microgr/mL.  I wonder if only a very small fraction of this drug would be absorbed.  As this seems a very high dose for this API and other poorly soluble APIs likely to be formulated as ASDs, it would be interesting to know the incidence of the pharmacobezoar at lower API/ASD doses.  Is this phenomenon only a concern at the very high doses shown here (1900+ mg/kg/day ASD), or does it manifest at lower doses?

To answer this question, we would like refer to the work in our previous study (cited as reference 4) and a work giving background on the tested NCE (reference 10). In reference 4, we presented the incidence of pharmacobezoars in all nonclinical toxicological studies and studies on embryofetal development conducted with exactly the same SD-ASD tested here in vivo. The lowest dose that induced pharmacobezoars was 571 mg/kg/day SD-ASD (13 and 26 weeks oral toxicity studies).

The key point in this study was however not the investigation of the dose dependency of pharmacobezoar formation. In order to assess the influence of the viscosity of the vehicle as primary study aim with a number of animals as low as possible to gain statistically significant results at a high probability, incidences had to be as high as possible without causing adverse effects during study duration.

The bioavailability of the NCE itself was no primary concern for us, as the tested dose was within the range of previously tested doses that induced no test item-related toxicologically relevant effects in rats. The reasons therefore are explained in line 335 – 337 and the reference 10 cited there.

Related, I could not tell if the authors or others had previously done MRI or other experiments on placebo HPMCAS particles prepared by spray drying or other techniques to look for the pharmacobezoar as a control experiment to determine whether the HPMCAS alone is largely responsible for the agglomerates. 

 The in vitro work previously reported by some of the present authors in ref. 9 also seems to have used similarly high-loaded ASDs of HPMCAS.  To what extent does the identity of the drug or the drug loading matter in formation of the agglomerates?   

The placebo HPMCAS experiment seems an important control to understand the cause of agglomerate formation, and to guide the readers who may want to design formulations for suspension dosing in rats for other drug-containing ASDs with HPMCAS.

Theoretically, the variety of polymers to prepare ASDs for preclinical testing is broad. As explained from line 57 to line 67, the pH‑dependent solubility can be expected to play an important role for the formation of pharmacobezoars as dissolved ASDs cannot form solid agglomerates. Besides the specific characteristics of HPMC-AS, pH-dependent soluble polymers are generally the key to prepare preferentially used suspensions of ASDs for preclinical testing.

Indeed, spray dried HPMC-AS has been tested in vivo and in vitro previously. As given in reference 4, spray dried HPMC-AS has been tested in doses up to 857 mg/kg/day to investigate possible correlating findings after SD-ASD - pharmacobezoars were found in previous studies. Spray dried HPMC-AS induced no pharmacobezoars. In vitro, it also obtained a lower agglomeration potential than the SD-ASD administered in this study. The dose tested in vivo matched the portion of HPMC-AS in the SD-ASD dose of the highest dose groups.

However, testing the agglomeration of spray dried HPMC-AS in this study was not relevant from our point of view as it is the combination of a poorly soluble drug and the pH-dependent polymer that is practically relevant in point of pharmacobezoar formation in nonclinical testing. Pure HPMC-AS will normally not be included in study groups and physicochemical properties of the SD-ASD of a poorly soluble, probably lipophilic drug and HPMC-AS will vary from that of spray dried HPMC-AS. Therefore, no correlations from the pharmacobezoar formation of spray dried HPMC-AS can be drawn on the formation of pharmacobezoars from HPMC-AS based SD-ASDs. Whereas results of the conducted study can be directly used to refine animal studies, a study in which pharmacobezoars should be formed from “placebo” HPMC-AS would probably be hardly justifiable for the explained reasons.

To investigate the role of the NCE for the pharmacobezoar formation in detail, we hope to get access to further SD-ASDs that formed pharmacobezoars in vivo by raising attention to this topic. In vitro testing of these would be interesting to investigate the role of the NCE and its drugload in the SD-ASD for pharmacobezoar formation. Based on that, we would be able to guide the preclinical formulation developers to develop formulations with low pharmacobezoar formation potential. At the moment, this is limited to the approach presented to reduce the potential if relevant.

From what we know so far, the pH dependent solubility of the applied polymer can be understood as a necessary but not sufficient criterion for pharmacobezoar formation.

The ASDs were dosed in an acidic vehicle.  Is this an important factor in causing the pharmacobezoar?  Many times HPMCAS-based ASDs can be dosed in a higher pH suspension vehicle.  Does this mitigate or completely prevent the formation of pharmacobezoar because the HPMCAS is more soluble at higher pH values?  This seems important to know.

To answer this question, we would like to refer to the context of the answer given before. If the pH of the vehicle enables dissolution of the suspended SD-ASD, unfavorable issues such as stability of the NCE, recrystallization and associated problems become relevant.

In the introduction, we pointed out the importance of the pH dependent solubility.

„In order to prevent SD-ASD from pharmacobezoar formation, an option would probably be the exchange of the polymer of SD-ASDs to a non-pH dependent soluble polymer, so that SD-ASDs dissolve at the latest during gastric residence time following oral administration. However, in addition to the outstanding potential of HPMC-AS to maintain supersaturation and inhibit recrystallization [8], it is particularly the pH‑dependent solubility that makes HPMC-AS a frequently chosen polymer for preclinical formulation of poorly soluble NCEs. Contrary to pH-independent soluble polymers, HPMC‑AS can be utilized to prepare suspensions of SD-ASDs in an acidic vehicle. These HPMC‑AS-based SD-ASD particles from orally administered suspensions would not dissolve prior to the rise of the environmental pH following gastric emptying into the duodenum. With dissolution to supersaturated solutions right at the place of absorption, optimal conditions for high bioavailability can be achieved. “ (line 56-67)

We also discussed the correlation of the pH of the vehicle and pKa of the polymer to its solubility in reference 4 and reference 9.

It was found in the previous in vitro study (cf. ref. 9), and in the present in vivo work, that adding HEC to the aqueous vehicle significantly mitigates the agglomerate formation.  The authors attribute this to increased viscosity of the HEC vehicle.  Maybe I missed it in the discussion, but it was not clear to me why a higher viscosity vehicle would mitigate agglomerate formation.  Is the mechanism known or obvious? 

Of course this is an important point. We discussed that in reference 9 and decided to cite this discussion here to avoid redundancies (“Referring to the physical background of agglomeration processes, it is known that enhanced particle motion and accompanying collisions enhance aggregation of particles[29]. Through the developed model able to study the effect of hindered particle motion due to viscosity enhancement of the vehicle, the significance of the effect was with a reduction of agglomerated mass by more than 85% impressive. “). Even though we cannot exclude that other processes such as interactions between HEC and the SD-ASDs also contribute to the effect, a detailed physical investigation underlying processes goes beyond the scope of this study and would be an interesting question for a scientist with a more physical background.

Assuming viscosity is an important factor, it would seem useful to know the viscosity of the suspensions used in the current manuscript.  What was the viscosity of the HCl vehicle vs. the HEC-based vehicle?  I assume the viscosity of a suspension having a lot of HPMCAS will depend on the pH.  Would the viscosity likely be significantly higher at higher pH for the suspensions reported here, i.e. 72 mg/mL HPMCAS-L?  Some more quantitative and detailed assessment of the viscosity would be valuable for a reader to understand and potentially use in design of suspension vehicles for dosing.

The 1% HEC vehicle obtained a viscosity of 2176 mPa*s at a shear rate of 1.29 * s-1 at 25°C using a Brookfield DV3T. However, the decision to not state numeric viscosity values was made deliberately for two reasons. First, for HEC containing fluids obtaining typically a pseudoplastic rheological behavior, the ratio of shear stress and shear rate is not linear. Thus, giving a numeric value of apparent viscosity might have the character of an in-process control rather than being a basis to compare both vehicles. Second, and maybe even more important, might the suspended SD-ASD influence the viscosity of the suspension. Thus, giving a concentration of a specific viscosity enhancer makes more sense to than giving numeric values to adjust vehicle or suspension to, that might furthermore have limited significance under in vivo conditions.

Reviewer 3 Report

Authors work is interesting and worth to publish in pharmaceutics.

Some minor comments should be addressed

1. Article did not follow author instrucution. like references etc.

2. if possible, author should do histopathalogy study of stomach.

Author Response

Answers to comments and suggestions of Reviewer 3

Dear reviewer, we appreciate your efforts in order to improve the manuscript and the valuable comments.

  1. Article did not follow author instrucution. like references etc.

Thank you for this comment, we corrected these points. This seems to have happened during resaving and renaming of the document with some references during the processing of the version to review for no obvious reason.

  1. if possible, author should do histopathalogy study of stomach.

Histopathology was not performed in this MRI study. For detailed discussion of induced Pathological findings including histopathology of the rodent stomach please see our first manuscript cited in thgis work as reference 4.

Reviewer 4 Report

1. Row 46 sounds bad as hydroxypropyl methylcellulose acetate succinate (HPMC-AS), it is more often a solid dissertation with hydroxypropyl methylcellulose acetate succinate. Think about it.

2. Line 47 is unclear who is in the acidified medium, the pH, and the reason. The terms carrier polymer is a bit confusing.

3. Although it is clear that the active substance is protected by a patent, some of its basic data could have been stated. Purpose, solubility, and does it have an absorption window?

4. Methanol is used as a spray-drying solvent. Do you have any information on whether it was tried with ethanol or a safer solvent? I think it is necessary to point out, although it is in the previous study, how much the increase in dissolution rate and solubility was.

5. 120 orders which apparent damage to animals

6. 119 line error in the reference

7. Table 2. on the basis of which you chose the vehicle for dissolving the formulation

8. 135, 184,252, 259, 276, 285, 317, 323, 371 errors in references... and there are more

9. Have you considered testing for acute and chronic nephro and hepatotoxicity? My problem is that I don't know what medicine it is and whether there is a need for it.

Author Response

Answers to comments and suggestions of Reviewer 4

Dear reviewer, we appreciate your efforts in order to improve the manuscript and the valuable comments.

  1. Row 46 sounds bad as hydroxypropyl methylcellulose acetate succinate (HPMC-AS), it is more often a solid dissertation with hydroxypropyl methylcellulose acetate succinate. Think about it.

Thank you for this comment. We reworded the corresponding sentence “In these studies, a spray dried amorphous solid dispersion (SD‑ASD) of the NCE BI 1026706 and hydroxypropyl methylcellulose acetate succinate (HPMC‑AS) was administered as suspension in an acidified aqueous vehicle [4].“

  1. Line 47 is unclear who is in the acidified medium, the pH, and the reason.

We assume that this comment refers to the vehicle used to prepare suspensions. In line 48, it is stated, that the SD-ASD is administered “suspended in acidified aqueous vehicle”. The importance of the pH-dependent solubility of HPMC-AS is discussed in line 55-66. For further information about the preclinical studies, including used vehicles, please see the cited reference 4.

The terms carrier polymer is a bit confusing.

The term has been previously used in this context in scientific work (e.g. see reference 8). For clarifictation, “carrier” was deleted.

  1. Although it is clear that the active substance is protected by a patent, some of its basic data could have been stated. Purpose, solubility, and does it have an absorption window?

From line 344-346 we stated “Pharmacological effects of the NCE (a bradykinin 1 receptor antagonist) were not expected as it already showed a low affinity and no test item-related toxicologically relevant effects in rats [10].“ Therefore, and due to the previous preclinical testing program conducted with this SD-ASD, bioavailability of the NCE was no critical point for this study. The solubility and other physicochemical properties are given in cited reference 9 and have not been repeated to avoid redundancies.

  1. Methanol is used as a spray-drying solvent. Do you have any information on whether it was tried with ethanol or a safer solvent? I think it is necessary to point out, although it is in the previous study, how much the increase in dissolution rate and solubility was.

Different solvents were tested during preclinical formulation and, of course, methanol is not the first choice if other solvents are similarly suitable, which was not the case for formulation 1. 

However, this formulation has been tested for residual moisture before administration, which was < 1%. Even if this 1% would have been exclusively methanol, which is unlikely due to methanol’s high vapor pressure, this concentration would be too low to induce adverse effects in rats during study duration.

Furthermore, it has to be considered that this was explicitly a preclinical formulation.

As enhancement of bioavailability by preparation of SD-ASDs is not the focus of this work, we would refrain discussion of numeric in vitro data.  SD-ASDs were prepared as administration crystalline suspensions did not result in adequate plasma levels of BI 1026706 to characterize the toxicological profile of the substance.  Administration of SD-ASD suspensions resulted in adequate plasma levels.  As stated in the manuscript and in the answer before, toxicological relevant effects were absent. 

  1. 120 orders which apparent damage to animals

Sadly, we were not able to identify what this comment refers to. We would kindly ask you to rephrase this comment if it is still relevant in the corrected manuscript.

  1. 119 line error in the reference
  2. 135, 184,252, 259, 276, 285, 317, 323, 371 errors in references... and there are more

Thank you for both comments. This seems to have happened during resaving and renaming of the document with some references during the processing of the version to review for no obvious reason, but has been corrected now.

  1. Table 2. on the basis of which you chose the vehicle for dissolving the formulation

The vehicle used to suspend the formulation was 0.01 N HCl in all groups of the dose finding study and Group A of the main study, as this vehicle lead to high incidences of pharmacobezoars in toxicological studies and high in vitro agglomeration. Please see reference 4 for more detailed information on the background of toxicological studies and reference 9 for background of in vitro testing.

  1. Have you considered testing for acute and chronic nephro and hepatotoxicity? My problem is that I don't know what medicine it is and whether there is a need for it.

As mentioned in answer to comment 3, tested SD-ASD was the same that had undergone extensive toxicological testing. No test item-related toxicologically relevant effects were observed in rats. Therefore, we focused investigations on the pharmacobezoar formation.

Round 2

Reviewer 2 Report

I thank the authors for their replies to the comments.  From what I understand, the authors chose not to incorporate any additional information or discussion into the manuscript based on my suggestions.  Please note that my initial suggestions were all aimed at enabling the reader to translate the very specific results here (for BI 1026706) into more general understanding that could be applied to other drugs and ASD formulations, which would make these results much more valuable to the pharmaceutical community, in my opinion.  I still believe this is the case.  Further brief reiteration of my suggestions to this end are below.  Thank you.

Dear reviewer, we appreciate your efforts in order to improve the manuscript and the valuable comments. You pointed out several questions that were also discussed during manuscript preparation. We discussed your comments (italic letters) in the following answers and look forward hearing back from you if there are still points that are not sufficiently explained from your point of view.

The doses studied are high. The authors stated their aim to use a “dose that entails a high incidence of pharmacobezoar without causing overt impairment of the animal’s general condition.” (lines 119-120) They used 2400 mg/kg/day of ASD for three weeks. At 70% drug load, this is 1680 mg/kg/day of a drug having an aqueous solubility of 20 microgr/mL. I wonder if only a very small fraction of this drug would be absorbed. As this seems a very high dose for this API and other poorly soluble APIs likely to be formulated as ASDs, it would be interesting to know the incidence of the pharmacobezoar at lower API/ASD doses. Is this phenomenon only a concern at the very high doses shown here (1900+ mg/kg/day ASD), or does it manifest at lower doses?

To answer this question, we would like refer to the work in our previous study (cited as reference 4) and a work giving background on the tested NCE (reference 10). In reference 4, we presented the incidence of pharmacobezoars in all nonclinical toxicological studies and studies on embryofetal development conducted with exactly the same SD-ASD tested here in vivo. The lowest dose that induced pharmacobezoars was 571 mg/kg/day SD-ASD (13 and 26 weeks oral toxicity studies).

The key point in this study was however not the investigation of the dose dependency of pharmacobezoar formation. In order to assess the influence of the viscosity of the vehicle as primary study aim with a number of animals as low as possible to gain statistically significant results at a high probability, incidences had to be as high as possible without causing adverse effects during study duration.

The bioavailability of the NCE itself was no primary concern for us, as the tested dose was within the range of previously tested doses that induced no test item-related toxicologically relevant effects in rats. The reasons therefore are explained in line 335 – 337 and the reference 10 cited there.

Thank you for the explanation.  I do understand that this is the same formulation tested in a preclinical study.  It is also the case, I believe, that this is a very high dose, so it might be good to at least restate from ref 4 the minimum dose for this formulation that caused the agglomerates, how many days of dosing it took, and in what percentage of animals.  Maybe this helps the reader put the current results into the context of dose as a starting point for assessing formulations of other drugs.

Second, if the purpose of the present study, as stated in the authors’ reply above, is to “assess the influence of the viscosity” on agglomerate formation, then it might be of value to state some quantitative measure of the viscosities of the formulations dosed.  It is difficult to assess the influence of viscosity if there is no definition or measure of viscosity.

Related, I could not tell if the authors or others had previously done MRI or other experiments on placebo HPMCAS particles prepared by spray drying or other techniques to look for the pharmacobezoar as a control experiment to determine whether the HPMCAS alone is largely responsible for the agglomerates.

The in vitro work previously reported by some of the present authors in ref. 9 also seems to have used similarly high-loaded ASDs of HPMCAS. To what extent does the identity of the drug or the drug loading matter in formation of the agglomerates?

The placebo HPMCAS experiment seems an important control to understand the cause of agglomerate formation, and to guide the readers who may want to design formulations for suspension dosing in rats for other drug-containing ASDs with HPMCAS.

Theoretically, the variety of polymers to prepare ASDs for preclinical testing is broad. As explained from line 57 to line 67, the pH‑dependent solubility can be expected to play an important role for the formation of pharmacobezoars as dissolved ASDs cannot form solid agglomerates. Besides the specific characteristics of HPMC-AS, pH-dependent soluble polymers are generally the key to prepare preferentially used suspensions of ASDs for preclinical testing.

Indeed, spray dried HPMC-AS has been tested in vivo and in vitro previously. As given in reference 4, spray dried HPMC-AS has been tested in doses up to 857 mg/kg/day to investigate possible correlating findings after SD-ASD - pharmacobezoars were found in previous studies. Spray dried HPMC-AS induced no pharmacobezoars. In vitro, it also obtained a lower agglomeration potential than the SD-ASD administered in this study. The dose tested in vivo matched the portion of HPMC-AS in the SD-ASD dose of the highest dose groups.

However, testing the agglomeration of spray dried HPMC-AS in this study was not relevant from our point of view as it is the combination of a poorly soluble drug and the pH-dependent polymer that is practically relevant in point of pharmacobezoar formation in nonclinical testing. Pure HPMC-AS will normally not be included in study groups and physicochemical properties of the SD-ASD of a poorly soluble, probably lipophilic drug and HPMC-AS will vary from that of spray dried HPMC-AS. Therefore, no correlations from the pharmacobezoar formation of spray dried HPMC-AS can be drawn on the formation of pharmacobezoars from HPMC-AS based SD-ASDs. Whereas results of the conducted study can be directly used to refine animal studies, a study in which pharmacobezoars should be formed from “placebo” HPMC-AS would probably be hardly justifiable for the explained reasons.

To investigate the role of the NCE for the pharmacobezoar formation in detail, we hope to get access to further SD-ASDs that formed pharmacobezoars in vivo by raising attention to this topic. In vitro testing of these would be interesting to investigate the role of the NCE and its drugload in the SD-ASD for pharmacobezoar formation. Based on that, we would be able to guide the preclinical formulation developers to develop formulations with low pharmacobezoar formation potential. At the moment, this is limited to the approach presented to reduce the potential if relevant.

From what we know so far, the pH dependent solubility of the applied polymer can be understood as a necessary but not sufficient criterion for pharmacobezoar formation.

My reason for asking about the placebo polymer results is that the typical journal reader likely has no particular interest in the specific NCE studied here.  Therefore, it would be valuable to translate the results here to other formulations based on either the amount of polymer or the drug properties.  If the authors believe the lack of agglomerates from HPMCAS placebo in their in vitro studies (reported previously) is representative of the in vivo situation, then that is a very important result to state here.  And if this is so, can anything else be said, even in speculation, about the type of drug, drug loading, etc. that causes pharmacobezoars?  This is likely what the reader would find most valuable.

The ASDs were dosed in an acidic vehicle. Is this an important factor in causing the pharmacobezoar? Many times HPMCAS-based ASDs can be dosed in a higher pH suspension vehicle. Does this mitigate or completely prevent the formation of pharmacobezoar because the HPMCAS is more soluble at higher pH values? This seems important to know.

To answer this question, we would like to refer to the context of the answer given before. If the pH of the vehicle enables dissolution of the suspended SD-ASD, unfavorable issues such as stability of the NCE, recrystallization and associated problems become relevant.

In the introduction, we pointed out the importance of the pH dependent solubility.

„In order to prevent SD-ASD from pharmacobezoar formation, an option would probably be the exchange of the polymer of SD-ASDs to a non-pH dependent soluble polymer, so that SD-ASDs dissolve at the latest during gastric residence time following oral administration. However, in addition to the outstanding potential of HPMC-AS to maintain supersaturation and inhibit recrystallization [8], it is particularly the pH‑dependent solubility that makes HPMC-AS a frequently chosen polymer for preclinical formulation of poorly soluble NCEs. Contrary to pH-independent soluble polymers, HPMC‑AS can be utilized to prepare suspensions of SD-ASDs in an acidic vehicle. These HPMC‑AS-based SD-ASD particles from orally administered suspensions would not dissolve prior to the rise of the environmental pH following gastric emptying into the duodenum. With dissolution to supersaturated solutions right at the place of absorption, optimal conditions for high bioavailability can be achieved. “ (line 56-67)

We also discussed the correlation of the pH of the vehicle and pKa of the polymer to its solubility in reference 4 and reference 9.

Thank you.  I am aware of the pH solubility profile of this and other polymers, and that sometimes, but not always, it can be helpful to dose HPMCAS and other enteric polymers at low pH.  An important question remains, do pharmacobezoars form if the formulation, or HPMCAS alone, is dosed in a higher pH vehicle?  It is not clear to me if there is in vitro or in vivo data from the current or previous studies that speak directly to this question.  It would useful to say something about this if possible.

It was found in the previous in vitro study (cf. ref. 9), and in the present in vivo work, that adding HEC to the aqueous vehicle significantly mitigates the agglomerate formation. The authors attribute this to increased viscosity of the HEC vehicle. Maybe I missed it in the discussion, but it was not clear to me why a higher viscosity vehicle would mitigate agglomerate formation. Is the mechanism known or obvious?

Of course this is an important point. We discussed that in reference 9 and decided to cite this discussion here to avoid redundancies (“Referring to the physical background of agglomeration processes, it is known that enhanced particle motion and accompanying collisions enhance aggregation of particles[29]. Through the developed model able to study the effect of hindered particle motion due to viscosity enhancement of the vehicle, the significance of the effect was with a reduction of agglomerated mass by more than 85% impressive. “). Even though we cannot exclude that other processes such as interactions between HEC and the SD-ASDs also contribute to the effect, a detailed physical investigation underlying processes goes beyond the scope of this study and would be an interesting question for a scientist with a more physical background.

Assuming viscosity is an important factor, it would seem useful to know the viscosity of the suspensions used in the current manuscript. What was the viscosity of the HCl vehicle vs. the HEC-based vehicle? I assume the viscosity of a suspension having a lot of HPMCAS will depend on the pH. Would the viscosity likely be significantly higher at higher pH for the suspensions reported here, i.e. 72 mg/mL HPMCAS-L? Some more quantitative and detailed assessment of the viscosity would be valuable for a reader to understand and potentially use in design of suspension vehicles for dosing.

The 1% HEC vehicle obtained a viscosity of 2176 mPa*s at a shear rate of 1.29 * s-1 at 25°C using a Brookfield DV3T. However, the decision to not state numeric viscosity values was made deliberately for two reasons. First, for HEC containing fluids obtaining typically a pseudoplastic rheological behavior, the ratio of shear stress and shear rate is not linear. Thus, giving a numeric value of apparent viscosity might have the character of an in-process control rather than being a basis to compare both vehicles. Second, and maybe even more important, might the suspended SD-ASD influence the viscosity of the suspension. Thus, giving a concentration of a specific viscosity enhancer makes more sense to than giving numeric values to adjust vehicle or suspension to, that might furthermore have limited significance under in vivo conditions.

As stated in response to the first question above, if the authors’ purpose of the study is to understand the effect of vehicle viscosity on pharmacobezoar formation, then it would seem useful to state some measure of the viscosity of the vehicles studied.  This may at least offer the reader some starting point for translating the results here to the design of other ASD suspensions.

Author Response

Dear Reviewer, thank you again for your comments. Please see the answers below (italic font) and the modifications of the manuscript regarding to your comments.

I thank the authors for their replies to the comments.  From what I understand, the authors chose not to incorporate any additional information or discussion into the manuscript based on my suggestions.  Please note that my initial suggestions were all aimed at enabling the reader to translate the very specific results here (for BI 1026706) into more general understanding that could be applied to other drugs and ASD formulations, which would make these results much more valuable to the pharmaceutical community, in my opinion.  I still believe this is the case.  Further brief reiteration of my suggestions to this end are below.  Thank you.

Thank you for the explanation.  I do understand that this is the same formulation tested in a preclinical study.  It is also the case, I believe, that this is a very high dose, so it might be good to at least restate from ref 4 the minimum dose for this formulation that caused the agglomerates, how many days of dosing it took, and in what percentage of animals.  Maybe this helps the reader put the current results into the context of dose as a starting point for assessing formulations of other drugs.

Following your suggestion, we added in line 57-60: „Observed incidences of pharmacobezoars positively correlated with dose and study duration. The lowest dose leading to pharmacobezoar formation was 571 mg SD-ASD administered once-daily for 13-weeks. “

Second, if the purpose of the present study, as stated in the authors’ reply above, is to “assess the influence of the viscosity” on agglomerate formation, then it might be of value to state some quantitative measure of the viscosities of the formulations dosed.  It is difficult to assess the influence of viscosity if there is no definition or measure of viscosity.

In the meantime, we conducted viscosity measurements that stated that there is no difference of in the shear rate dependent viscosity of the high-viscous vehicle and the viscosity data given in the manufacturers brochure for solutions of 1% Natrosol in water. Therefore, we added following statement to the manuscript in line 113-116: „The high-viscous vehicle was found to obtain the same shear thinning behavior that is described in the manufacturer brochure for solutions of 1% Natrosol® 250HX in water with a viscosity of 10 Pa*s at a shear rate of 0.01/s to 0.1 Pa*s at a shear rate of 1000/s [10].”

My reason for asking about the placebo polymer results is that the typical journal reader likely has no particular interest in the specific NCE studied here.  Therefore, it would be valuable to translate the results here to other formulations based on either the amount of polymer or the drug properties.  If the authors believe the lack of agglomerates from HPMCAS placebo in their in vitro studies (reported previously) is representative of the in vivo situation, then that is a very important result to state here.  And if this is so, can anything else be said, even in speculation, about the type of drug, drug loading, etc. that causes pharmacobezoars?  This is likely what the reader would find most valuable.

To clarify that pharmacobezoars were not observed from pure spray dried HPMC-AS, we added following statement to the introduction in line 60-62: “Pharmacobezoars were not observed following administration of pure spray-dried HPMC-AS in doses equal to the share of polymer on the SD‑ASD dose administered to the high dose groups [4].” We agree, that a deeper understanding of parameters of the SD-ASDs that determine the pharmacobezoar formation potential would be desirable. This will subject of further studies if we get access to more SD-ASDs that formed pharmacobezoars in vivo or obtain a high in vitro agglomeration potential on the level of the tested SD-ASD.

Thank you.  I am aware of the pH solubility profile of this and other polymers, and that sometimes, but not always, it can be helpful to dose HPMCAS and other enteric polymers at low pH.  An important question remains, do pharmacobezoars form if the formulation, or HPMCAS alone, is dosed in a higher pH vehicle?  It is not clear to me if there is in vitro or in vivo data from the current or previous studies that speak directly to this question.  It would useful to say something about this if possible.

To the best of our knowledge, there is no relevant in vivo data referring to this question as the tested SD-ASD of BI1026706 and HPMC‑AS was administered to rats suspended in an acidic vehicle. As we would expect that at least a certain share of the suspended SD-ASD dose would dissolve in a vehicle with a pH higher than 5.5, we have also not tested higher pH vehicles in vitro.

As stated in response to the first question above, if the authors’ purpose of the study is to understand the effect of vehicle viscosity on pharmacobezoar formation, then it would seem useful to state some measure of the viscosity of the vehicles studied.  This may at least offer the reader some starting point for translating the results here to the design of other ASD suspensions.

As mentioned in the answer to the second question, we conducted viscosity measurements, included viscosity values of the vehicle in the manuscript and cited the manufacturers brochure that contains detailed information on rheological characteristics regarding viscosity enhancement by HEC.

Reviewer 4 Report

.

Author Response

(The authors gave the same response as above.)

Round 3

Reviewer 2 Report

I thank the authors for their responses to the comments.